
# Measurement of $^3$He analyzing power for $p-^3$He scattering using the polarized $^3$He target

Atomu Watanabe[1*], Shinnosuke Nakai[1], Kimiko Sekiguchi[1], Tomomi Akieda[1], Daijiro Etoh[1], Minami Inoue[1], Yoshinori Inoue[1], Kenta Kawahara[1], Hiroshi Kon[1], Kenjiro Miki[1], Tomoyuki Mukai[1], Daisuke Sakai[1], Shun Shibuya[1], Yuta Shiokawa[1], Takahiro Taguchi[1], Hiroo Umetsu[1], Yuta Utsuki[1], Yasunori Wada[1], Morihiro Watanabe[1], Masatoshi Itoh[2], Takashi Ino[3], Takashi Wakui[4], Kichiji Hatanaka[5], Hiroki Kanda[5], Hooi Jin Ong[5], Dinh Trong Tran[5], Shuhei Goto[6], Yuma Hirai[6], Daiki Inomoto[6], Hina Kasahara[6], Shinji Mitsumoto[6], Hisanori Oshiro[6], Tomotsugu Wakasa[6], Yukie Maeda[7], Kotaro Nonaka[7], Hideyuki Sakai[8] and Tomohiro Uesaka[8]

1 Department of Physics, Tohoku University, Sendai, Miyagi 980–8578, Japan
2 Cyclotron and Radioisotope Center (CYRIC),
Tohoku University, Sendai, Miyagi 980–8578, Japan
3 High Energy Accelerator Research Organization (KEK), Tsukuba, Ibaraki 305–0801, Japan
4 National Institute of Radiological Science, Chiba 263–8555, Japan
5 Research Center for Nuclear Physics (RCNP),
Osaka University, Ibaraki, Osaka 567–0047, Japan
6 Department of Physics, Kyushu University, Fukuoka 812–8581, Japan
7 Faculty of Engineering, University of Miyazaki, Miyazaki 889–2192, Japan
8 RIKEN Nishina Center, Wako, Saitama 351–0198, Japan

⋆ watanabe@lambda.phys.tohoku.ac.jp

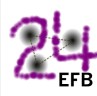
## Abstract

Proton–$^3$He scattering is one of the good probes to study the $T = 3/2$ channel of three–nucleon forces. We have measured $^3$He analyzing powers for $p-^3$He elastic scattering with the polarized $^3$He target at 70 and 100 MeV. In the conference the data were compared with the theoretical predictions based on the modern nucleon–nucleon potentials. Large discrepancies were found between the data and the calculations at the angles where the $^3$He analyzing power takes the minimum and maximum values.

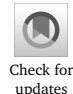
## 1 Introduction

Understanding nuclear properties based on bare nuclear forces is one of the main interests in nuclear physics. Three–nucleon forces (3NFs) are essentially important to clarify various nu-

clear phenomena such as few–nucleon scattering [1], binding energies of light mass nuclei [2] and equation of state of nuclear matter [3]. To investigate dynamical aspects (momentum, spin and isospin dependence) of 3NFs, the few-nucleon scattering is a good probe. One can perform direct comparison between the rigorous numerical calculations and precise experimental data, and extract the information of 3NFs. With the aim of studying 3NFs, we have performed the experimental studies of nucleon–deuteron scattering at intermediate energies (65−300 MeV/nucleon) [4,5]. In the case of the cross section for the elastic deuteron–proton scattering at 135 MeV/nucleon, large discrepancies between the experimental data and the rigorous numerical calculations based on the modern nucleon–nucleon ($NN$) potentials (such as AV18 [6], CD–Bonn [7] and Nijmegen I, II, 93 [8]) are found. The theoretical calculations with the Tucson-Melbourne'99 [9] or Urbana IX [10] 3NF reproduce well the cross section data. This is taken as the first signature of 3NF effects in three-nucleon scattering.

Recently, the importance of the isospin dependence of 3NFs has been suggested for understanding of asymmetric nuclear matter (e.g., neutron-rich nuclei and neutron matter properties) [2,3]. However, the total isospin of $Nd$ scattering system is limited to $T = 1/2$. In order to explore the properties of the 3NFs in $A \geq 3$ nuclear systems and to approach the isospin dependence of 3NFs, we have performed the measurement of $p$–$^3$He elastic scattering at intermediate energies. In this contribution we report the measurement of $^3$He analyzing powers for $p$–$^3$He elastic scattering at CYRIC, Tohoku University and RCNP, Osaka University using the polarized $^3$He target.

In this paper, we present the measurement of $^3$He analyzing powers at 70 and 100 MeV using the polarized $^3$He target. Section 2 deals with the details of the polarized $^3$He target system developed for $p$–$^3$He scattering experiment. In Sec. 3 we describe the measurement of $^3$He analyzing powers, and Section 4 presents comparison between the obtained data and the theoretical predictions. We summarize and conclude in Sec. 5.

## 2   Polarized $^3$He Target

We developed the polarized $^3$He target system for the measurement of $^3$He analyzing power. A schematic view of the polarized $^3$He target system is shown in Fig. 1. The polarized $^3$He target system consisted of main coils which were the Helmholtz type providing a static magnetic field, and coils for polarimetry, the oven and the laser system.

In order to polarize $^3$He nucleus, we adopted the alkali–hybrid spin–exchange optical pumping (AH–SEOP) method. This method consisted of two step processes. First, Rb vapor was polarized by optical pumping with circularly polarized light in the presence of a static magnetic field. Then the polarization of Rb atoms was transferred to K atoms via spin–exchange collisions. Second, the alkali metal polarization was transferred to $^3$He nuclei by hyper–fine interactions. We used a spectrally narrowed laser with the optics to produce circularly polarized light. The output power of the laser was 60 W and the wavelength was 795 nm with FWHM of 0.2 nm. The target cell consisted of a cylindrical pumping chamber with a length of 45 mm and a cylindrical target chamber with a length of 150 mm. A diameter of the pumping chamber and the target chamber were 60 mm and 40 mm, respectively. It contained the $^3$He gas with pressure of 3 atm. at room temperature together with a small amount of Rb and K as well as $N_2$ gas. The target cell was installed at the center of the main coils, and the pumping chamber was placed in the oven to obtain a sufficient amount of Rb and K vapor by heating up to $\sim 200$ °C.

We measured the target polarization by the adiabatic fast passage–NMR (AFP–NMR) method in combination with the electron paramagnetic resonance (EPR) method [11]. The AFP–NMR system consisted of the main coils, drive coils and a pick-up coil. Sweeping a static

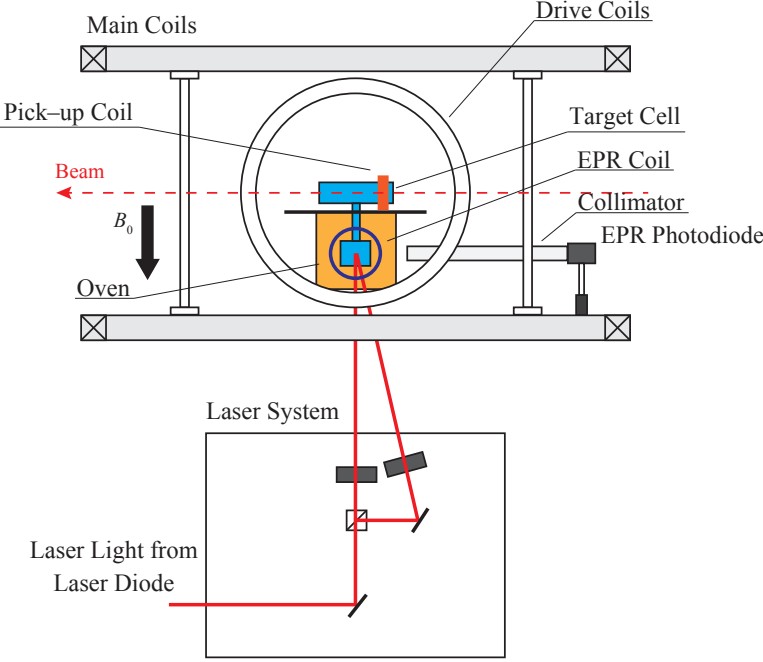

Figure 1: (Color online) Schematic view of the polarized $^3$He target.

magnetic field while applying a RF field by the drive coils under the AFP condition, we flipped the direction of $^3$He spin. The induced NMR signals were detected by the pick–up coil. We have obtained the absolute values of the $^3$He polarization by the EPR method. This method uses the EPR frequency shift due to the magnetic filed created by polarized $^3$He nuclei. The EPR system consisted of a EPR coil placed inside the oven and a photodiode to detect a EPR signal. The EPR coil created an RF magnetic filed to induce EPR. However, the EPR measurement gives us only the $^3$He polarization of the pumping chamber because it is needed to the mixture of $^3$He and alkali metal vapor. It is necessary to know the $^3$He polarization of the target chamber to extract $^3$He analyzing powers from the $p-^3$He elastic scattering experiment. For this reason, we have performed the neutron transmission measurement which offers direct measurement of the $^3$He polarization in the target chamber. Neutron transmission $T_n$ is described as;

$$T_n = e^{-\sigma_{\mathrm{abs}} n_{\mathrm{He}} d} \cosh(P_{\mathrm{He}} \sigma_{\mathrm{abs}} n_{\mathrm{He}} d), \tag{1}$$

where $\sigma_{\mathrm{abs}}$ is the neutron absorption cross section of $^3$He, $n_{\mathrm{He}}$ is the $^3$He number density, $d$ is the inner length of the target chamber and $P_{\mathrm{He}}$ is the $^3$He polarization, respectively. Therefore, the $^3$He polarization is calculated as,

$$P_{\mathrm{He}} = -\frac{1}{\ln T_{n,0}} \cosh^{-1}\left(\frac{T_n}{T_{n,0}}\right), \tag{2}$$

where $T_{n,0}$ is the neutron transmission with the un–polarized $^3$He cell. The experiment has been performed using a neutron source at RIKEN. We obtained the $^3$He polarization of the target to be about 40 %.

## 3   Detailed Measurement of $^3$He Analyzing Powers

The polarized cross section for $p-^3$He elastic scattering is expressed as,

$$\frac{d\sigma}{d\Omega} = \left(\frac{d\sigma}{d\Omega}\right)_0 (1 + p_y A_y),\tag{3}$$

where $d\sigma/d\Omega$ $((d\sigma/d\Omega)_0)$ is the polarized (un–polarized) differential cross section, and $p_y$ denotes the polarization of the $^3$He target. The polarization axis of the target is normal to the reaction plane in this experiment. The $^3$He analyzing power can be extracted by applying the polarized $^3$He target with the upward and downward directions.

We performed the measurement of $^3$He analyzing powers for $p-^3$He elastic scattering at CYRIC, Tohoku University and RCNP, Osaka University. The experimental conditions are summarized in Table 1.

Table 1: Summary of the experimental conditions.

| Facilities | CYRIC | RCNP |
|---|---|---|
| Observables | $A_{0y}$ | $A_{0y}$ |
| Beam | proton | proton |
| Energy | 70 MeV | 100 MeV |
| Beam intensity | $5-10$ nA | 30 nA |
| Target | $^3$He gas (2 mg/cm$^2$) | $^3$He gas (2 mg/cm$^2$) |
| Target polarization | max. $\sim 40$ % | max. $\sim 40$ % |
| Detectors | $\Delta E$–$E$ detectors | $\Delta E$–$E$ detectors |
| Measured angles | $\theta_{\text{c.m.}} = 46° - 141°$ | $\theta_{\text{c.m.}} = 47° - 149°$ |
| Solid angles | 0.4 msr | 0.4 msr |

A schematic view of the experimental setup at CYRIC is shown in Fig. 2. The proton beams were accelerated up to 70 MeV by the AVF cyclotron. Typical beam intensities were 5 – 10 nA. Charge collection of the beams was performed by using the Faraday cup placed downstream of the target. We also monitored relative values of the beam intensities by the beam monitor installed in the vacuum chamber placed upstream of the target. The scattered protons from the polyethylene film with a thickness of 20 $\mu$m were detected by the beam monitor at $\theta_{\text{lab.}} = 45°$. The beam monitor was $\Delta E - E$ detectors which consists of thin and thick plastic scintillators coupled with photomultiplier tubes (PMTs). The scattered protons from the polarized $^3$He target were detected by $\Delta E - E$ detectors placed on a left and right side of the target. The $\Delta E$ detectors were plastic scintillators with thickness of 0.2, 0.5 or 1 mm. The $E$ detectors were a NaI(Tl) scintillator with thickness of 50 mm. The measured angles were $\theta_{\text{c.m.}} = 46° - 141°$ in the center of mass system. During the experiment, we measured the $^3$He polarization and flipped $^3$He nuclear spin direction by using the AFP–NMR method every hour.

We also performed the measurement of $^3$He analyzing powers at RCNP, Osaka University. The proton beams were provided by a ion source and then were accelerated up to 100 MeV by the AVF cyclotron and the Ring cyclotron. A typical beam intensity was $\sim 30$ nA. Charge collection of the beams was performed by using the Faraday cup placed downstream of the target. An experimental setup around the target was the same as that of CYRIC. The measured angles were $\theta_{\text{c.m.}} = 47° - 149°$ in the center of mass system.

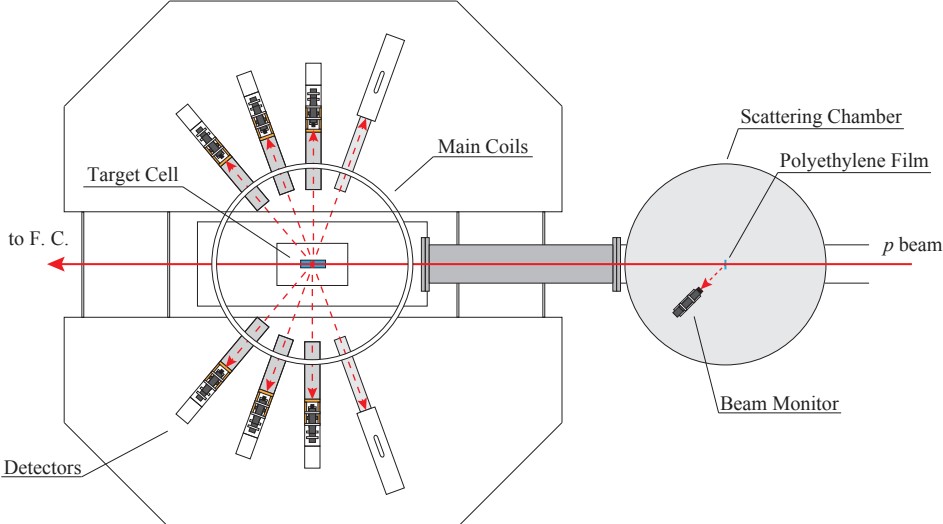

Figure 2: (Color online) Schematic view of the experimental setup for $p-^3$He scattering at CYRIC.

## 4   Results

Figure 3 shows the experimental data for the $^3$He analyzing powers. In the conference the experimental data were compared with the rigorous numerical calculations based on various realistic nuclear potentials [12]. The angular distribution of the experimental data had a moderate agreement with the theoretical predictions. However, the large discrepancies were found at around minimum and maximum angles both at 70 and 100 MeV. Careful analysis is in progress now together with the evaluation of the target $^3$He polarization.

## 5   Summary and Conclusions

We have performed the measurement of $^3$He analyzing powers for $p-^3$He elastic scattering at 70 and 100 MeV by using the polarized $^3$He target in a wide angular range. In the conference the experimental data were compared with the rigorous numerical calculations based on the modern $NN$ potentials. For both the incident energies large discrepancies were found between the data and the calculations at around the angles where the data takes maximum and minimum values. For the lower energies 5–35 MeV, the data of $^3$He analyzing powers are reproduced by the theoretical predictions based on the $NN$ potential only [13, 14]. The obtained results indicate that some other components are missed in the theoretical prediction in $p-^3$He scattering. In order to perform detailed quantitative discussions of the 3NF effects, study is in progress both from experiment and theory.

## Acknowledgements

We performed the scattering experiments at CYRIC, Tohoku University and RCNP, Osaka University. We also performed the measurement of the target polarization at RANS, RIKEN. We acknowledge all the accelerator groups for providing high quality beams. This work was supported financially in part by the Grants-in-Aid for Scientific Research No. 25105502, No.

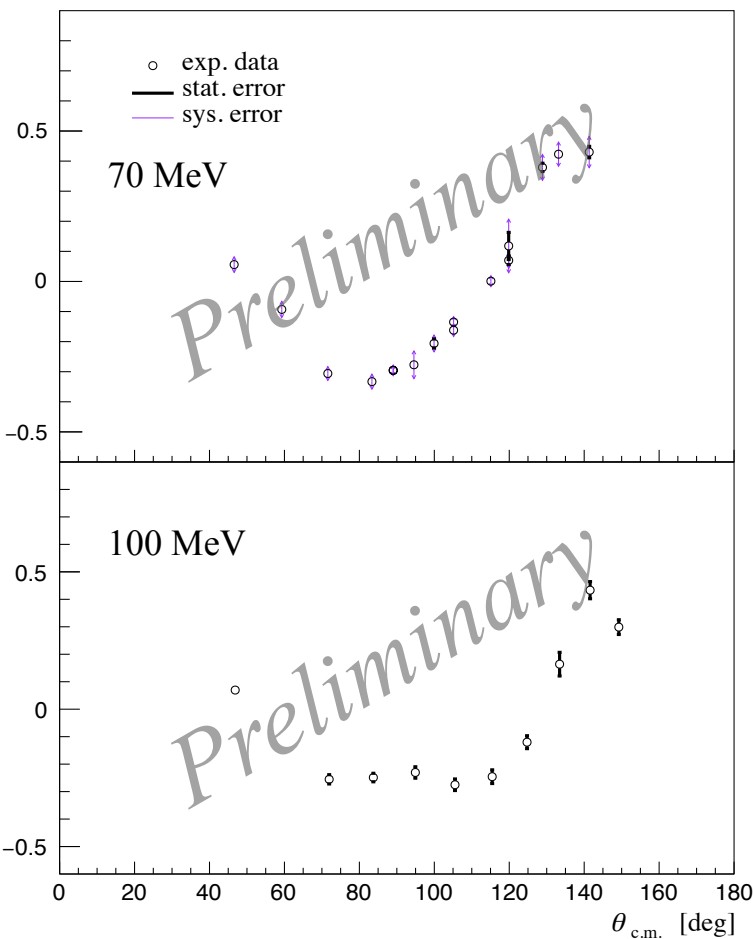

Figure 3: (Color online) $^3$He analyzing powers for $p-^3$He elastic scattering at 70 and 100 MeV. The theoretical calculations are not shown here.

16H02171, and No. 18H05404 of the Ministry of Education, Culture, Sports, Science, and Technology of Japan.

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
