# Peer review of "Measurement of 3He analyzing power for p-3He scattering using the polarized 3He target"

_SciPost Physics Proceedings, doi:SciPost Phys. Proc. 3, 020 (2020)_

## Round 1 · Referee Report · Anonymous (Referee 1) · 2019-12-2

Report

This proceedings is very nicely written. The presented work is very interesting and up to date. I wait for more results from RCNP on p-3He scattering.
From my point of view it would be very interesting to see the plot of polarization changing in the time of the experiment. I would also appreciate if the authors
estimated the (systematic) uncertainty of the measured polarization (40%+/-...) or to present the approach to establish such error. How accurate is you method ?

Requested changes

1- Introduction: "Three–nucleon forces (3NFs) are essentially important to clarify various nuclear properties such as few–nucleon scattering." maybe better word will be "phenomena" instead of "properties". 2- at the end of the 3'rd section: "A experimental setup" -> "An experimental setup" 3- Results: "... the theoretical prediction of the CD–Bonn potential with ∆ degrees of freedom" maybe more precise: "... the theoretical predictions of the CD–Bonn potential with ∆-isobar degrees of freedom included".

  • validity: high
  • significance: high
  • originality: high
  • clarity: high
  • formatting: excellent
  • grammar: good

Author:  Atomu Watanabe  on 2019-12-16  [id 681]

(in reply to Report 1 on 2019-12-02)

Thank you very much for your refereeing and comments.
I have done resubmission my contribution reflected your requests.

The 3He polarization was stable during the experiments.
In the experiments for about two days, the difference of the 3He polarization between the average and maximum values was about 3 % in magnitudes.
The (statistic) uncertainty of the polarization is about 3 % by the method of neutron transmission measurements.
The systematic uncertainty of this method (mainly comes from beam intensity uncertainties) is rather smaller than the statistic one.

---

## Editorial Decision

published